# 4K-memristor analog-grade passive crossbar circuit

H. Kim [1,2], M. R. Mahmoodi [1], H. Nili [1] & D. B. Strukov [1✉]

The superior density of passive analog-grade memristive crossbar circuits enables storing large neural network models directly on specialized neuromorphic chips to avoid costly off-chip communication. To ensure efficient use of such circuits in neuromorphic systems, memristor variations must be substantially lower than those of active memory devices. Here we report a 64 × 64 passive crossbar circuit with ~99% functional nonvolatile metal-oxide memristors. The fabrication technology is based on a foundry-compatible process with etch-down patterning and a low-temperature budget. The achieved <26% coefficient of variance in memristor switching voltages is sufficient for programming a 4K-pixel gray-scale pattern with a <4% relative tuning error on average. Analog properties are also successfully verified via experimental demonstration of a 64 × 10 vector-by-matrix multiplication with an average 1% relative conductance import accuracy to model the MNIST image classification by ex-situ trained single-layer perceptron, and modeling of a large-scale multilayer perceptron classifier based on more advanced conductance tuning algorithm.

[1] Department of Electrical and Computer Engineering, University of California, Santa Barbara, CA, USA. [2] Present address: Department of Electronic Engineering of Inha University, Incheon, Korea. ✉email: strukov@ece.ucsb.edu

A nalog-grade nonvolatile memories, such as those based on floating-gate transistor[1–3], phase-change[4–6], ferroelectric[7,8], magnetic[9], solid-state electrolyte[10–13], organic[14,15], and metal-oxide[16–28] materials, are enabling components for mixed-signal circuits implementing vector-by-matrix multiplication (VMM), which is the most common operation in any artificial neural network. Most importantly, such circuits allow for physical-level in-memory computations in the analog domain using the fundamental Ohm and Kirchhoff laws, thus enabling dramatically higher energy and area efficiency in comparison with digital solutions. The main advantages of using passively integrated metal-oxide memristors[17,22,24], which are also referred to as resistive random-access memories (ReRAMs), are their superior density and lower fabrication cost[29]. In fact, due to excellent scaling prospects and analog properties, vertically integrated ReRAMs might challenge much slower 3D NAND memories in effective density to enable human-brain-scale integrated electronics.

There has been substantial progress in the development of 1T-1R memory arrays, in which a memory cell based on a two-terminal resistive switching element (1R) also includes one dedicated select transistor (1T), and numerous demonstrations from academia and industry of using such active memories in neuromorphic computing circuits—see e.g. refs., [18–20,23,25–27], and also recent reviews[30–34]. Perhaps, the most impressive neuromorphic functionality was reported based on nonvolatile TaO$_{2-x}$ devices integrated into $128 \times 64$ active crossbar arrays—see details of such devices in ref. [23] and review of many experimental demonstrations based on such technology in refs. [33]. The main weakness of that technology, however, is extremely large, of the order of 2500 μm$^2$, size of 1T1R cell, and high (mS-scale) device conductance, which necessitates bulky and energy-hungry peripheral circuits. In addition, the reported excellent conductance tuning results are partly due to the use of the select transistor in 1T1R cell, which inhibits half-select disturbance—the main challenge for achieving high precision tuning in passively integrated circuits (Supplementary Fig. 1).

The progress in the most prospective, passive analog-grade ReRAM circuits[13,17,21,22,24], however, has been much slower, mainly because of much stricter requirements for the uniformity of memory cells' $I–V$ characteristics (Supplementary Fig. 1). For example, Xpoint memory—the most advanced commercialized technology using passively integrated memory devices—operates in a digital mode. (Such memory is also most likely based on phase-change materials[35], which are less appealing for analog computing applications due to larger conductance drift over time.) A promising $I–V$ uniformity results with very tight variations were reported for stand-alone devices based on organic[15] and epitaxial[11] materials. The main concern for these recently developed analog-grade memristors is the compatibility of the utilized fabrication flows with conventional semiconductor foundry processes. Reference [21] describes 500-nm half-pitch $32 \times 32$ circuits based on W/WO$_x$/Pd/Au devices, which were tuned with 25% precision (estimated from Fig. S3d data) to implement a sparse encoding algorithm. A similar device technology was recently used by the same group to demonstrate large-scale fully integrated complementary metal-oxide-semiconductor (CMOS)/memristor circuits[25]. It is not clear, however, if the reported results in ref. [25] were obtained based on reading conductances after completing the tuning process for all devices in the crossbar circuit or just a fraction of them, as it was performed by the same authors in ref. [12]. An even more serious and related concern is a lack of detailed statistics and, most importantly, data on retention because similar devices were shown to be volatile due to interfacial switching mechanisms according to the previous studies[36]. Another very recent work reported analog-grade

$32 \times 32$ crossbar arrays based on passively integrated Si-alloy:Ag electrochemical devices[13]. Though a very impressive 100% device yield and highly linear state update characteristics were reported, the main weakness of that work is also poor retention of the devices. Additional concerns are whether the yield results reported for $10 \times 10$ μm$^2$ footprint crosspoint devices will hold for nanoscale devices and the use of silver in the device stack, a contaminant typically avoided in CMOS foundry processes. Reference [28] proposed a very promising concept for a three-dimensional memristive memory. Unfortunately, all presented experimental results in that paper were obtained for a rather unpractical structure based on microscale binary-switching devices with non-overlapping footprints so that the demonstrated three-dimensional integration does not improve the effective memristor density.

Supplementary Table 1 summarizes experimental work on analog-grade 1T1R and 0T1R metal-oxide memristor crossbars. As evident from this table, the uniformity, density, and analog properties of previously reported memristive crossbar circuits are insufficient for making practical neuromorphic hardware, especially for running large-scale neural models. The main contribution of this work is to address these challenges by developing uniform CMOS-compatible fabrication technology for building larger, conducive for back-end-of-the-line 3D integration crossbar array circuits and showing the prospects of such technology in neuromorphic computing applications. The developed circuits have ten times more devices and excellent uniformity allowing for significantly better array-scale conductance tuning precision as compared to the previous work[24] that reported the largest passive analog-grade memristive crossbar circuits with detailed characterization statistics. Moreover, the demonstrated artificial neural network is close in complexity to the state-of-the-art neuromorphic prototypes based on (>10,000 sparser and 10× more conductive) 1T1R ReRAM devices[23,33].

## Results

**Device fabrication.** The developed $64 \times 64$ crossbar circuit consists of Ti/Al/TiN-based top and bottom electrodes and an Al$_2$O$_3$/TiO$_{2-x}$ switching layer (Fig. 1). The actual crossbar array dimensions are $(64 + 2) \times (64 + 2)$, with an additional line added at both sides of the circuit for the top and bottom layers to achieve better uniformity for the devices in the main array. The bottom and top electrodes and titanium oxide layers are deposited by reactive sputtering, while aluminum oxide is formed with an atomic layer deposition (ALD) technique. The bottom electrode is planarized via a combination of chemical-mechanical polishing (CMP) and etch-back. All crossbar circuit features are patterned using photolithography and etching process—see "Methods" sections for more fabrication details.

Though the developed technology builds upon the previous work on Al$_2$O$_3$/TiO$_{2-x}$ devices, several essential improvements enable the demonstration of functional larger-scale crossbar circuits. The similarities are, for example, in that the thin titanium layer in the electrodes provided adhesion, and, in the case of the top electrodes, is used to create an ohmic interface with large oxygen vacancy concentration near the top portion titanium oxide film[17]. Instead of relying on precisely controlling stoichiometry during deposition[37], we have opted for thermal annealing to adjust the oxygen vacancy profile, which results in the gettering of top electrode titanium metal and diffusion of oxygen vacancies towards the bottom interface[17]. Such fine-tuning of oxygen vacancy doping allows lowering conductances of as-fabricated memristors and hence reducing voltages for the device forming (and eliminating the forming step for some), and is necessary for crossbar integration and improving device

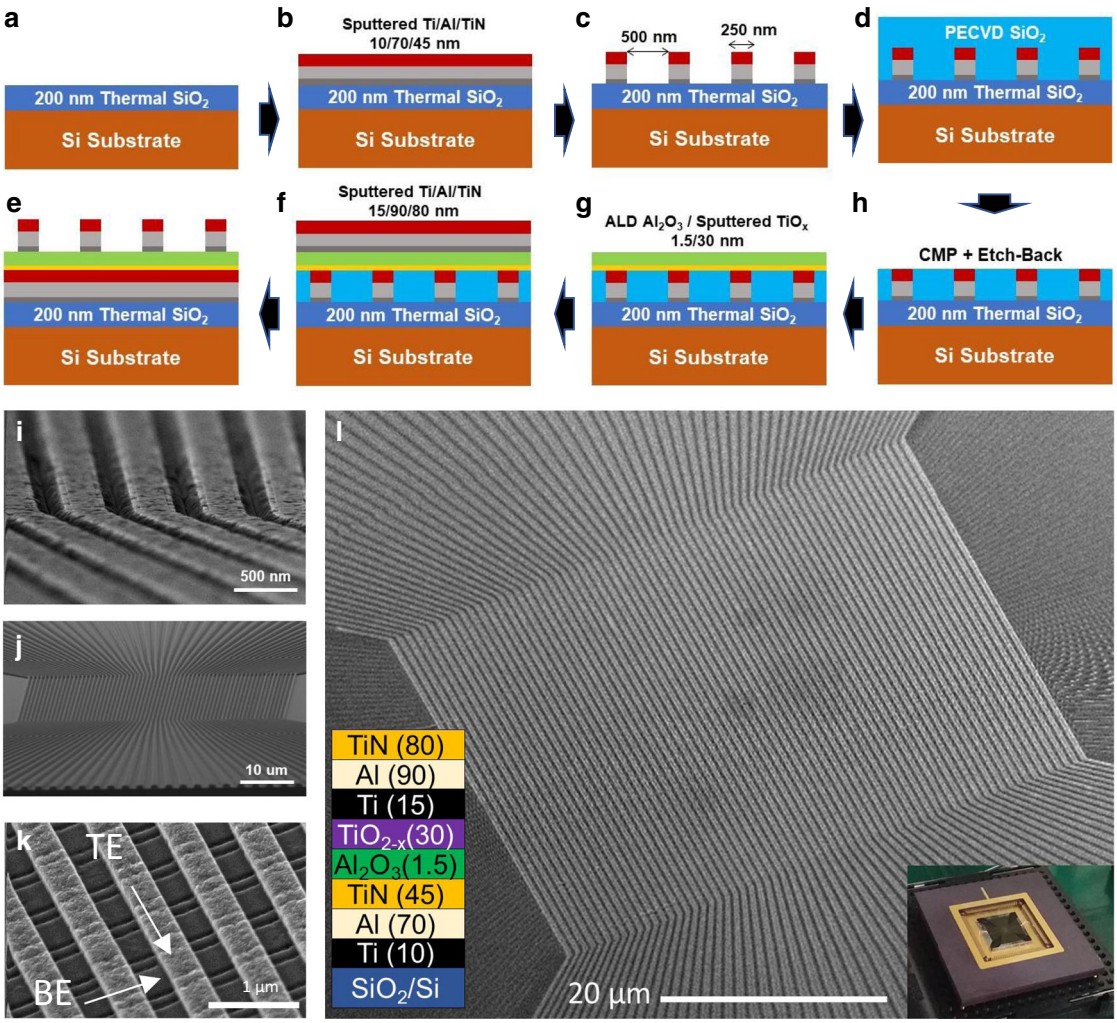

**Fig. 1 Device fabrication details. a–h** Growth and patterning process steps (see "Methods" section for details). For clarity, panel e shows device cross-section turned out-of-plane by 90 degrees with respect to drawings shown in panels **a–d**, **f–h**. Scanning electron microscopy (SEM) images of (**i**) patterned bottom electrodes, (**j**) partially planarized bottom electrodes through chemical-mechanical polishing and etch-back, and (**k**) a fragment of completed crossbar array. **l** SEM image of the full 64 × 64 memristor crossbar array. Bottom left and bottom right insets show, correspondingly, material layers at the device cross-section with corresponding thicknesses in nanometers, and the packaged chip.

uniformity. The aluminum oxide layer, with parameters optimized similarly to ref. [17], is integrated into the stack to suppress device currents at small voltages and bottom-to-bottom line leakages and hence improve the dynamic on-off range.

Let us stress the importance of several distinctive techniques that improve line conductances, device uniformity, and yield and are essential for scaling up the crossbar size. First, aluminum is selected for its better conductivity, instead of commonly used noble and other higher-resistance inert materials in other works[17,21,22,24]. The inert titanium nitride capping is needed to avoid aluminum oxidation. Second, patterning via reactive ion etching, instead of the typically employed lift-off process, allows fabricating larger (>1/2) aspect-ratio electrodes. It also improves the quality of top electrodes, e.g., by eliminating the undesirable formation of kinks at line edges ("rabbit ears"), which are typical for the lift-off patterning. It also helps avoid sidewall residue along bottom line edges (Supplementary Fig. 2), which is similar to gate spacer residue at the Si fin channel during the FinFET process flow[38]. Ultimately, the combination of etch-back and properly calibrated CMP processes (Supplementary Fig. 3) ensures better step coverage. (On the other hand, planarization by CMP process only was found to cause significant damage to

the surface of bottom electrodes). It is worth mentioning that ion milling and CMP techniques have been previously used to fabricate 2 × 10 × 10 crossbar circuits[22]; however, the primary purpose of these techniques was to enable vertical monolithic integration, while, e.g., line resistance remained large due to the use of small-aspect ratio Pt electrodes. In addition, there are other essential differences in the calibration of the planarization step (see "Methods" section).

Scanning electron microscopy images of the fabricated crossbar array (Fig. 1i–l) confirm the top electrodes' smooth planar topology and their structural isolation with no noticeable sidewall residue between them. With all the modifications, the developed fabrication process has a low-temperature fabrication budget. It can be adopted by silicon foundries for back-end-of-line integration and vertical monolithic integration of multiple crossbar arrays.

**Device characterization.** Current-voltage characteristics for the as-fabricated devices, i.e., before applying the electroforming process, are fairly uniform (Supplementary Fig. 4), which is an essential prerequisite for lowering variations in functional

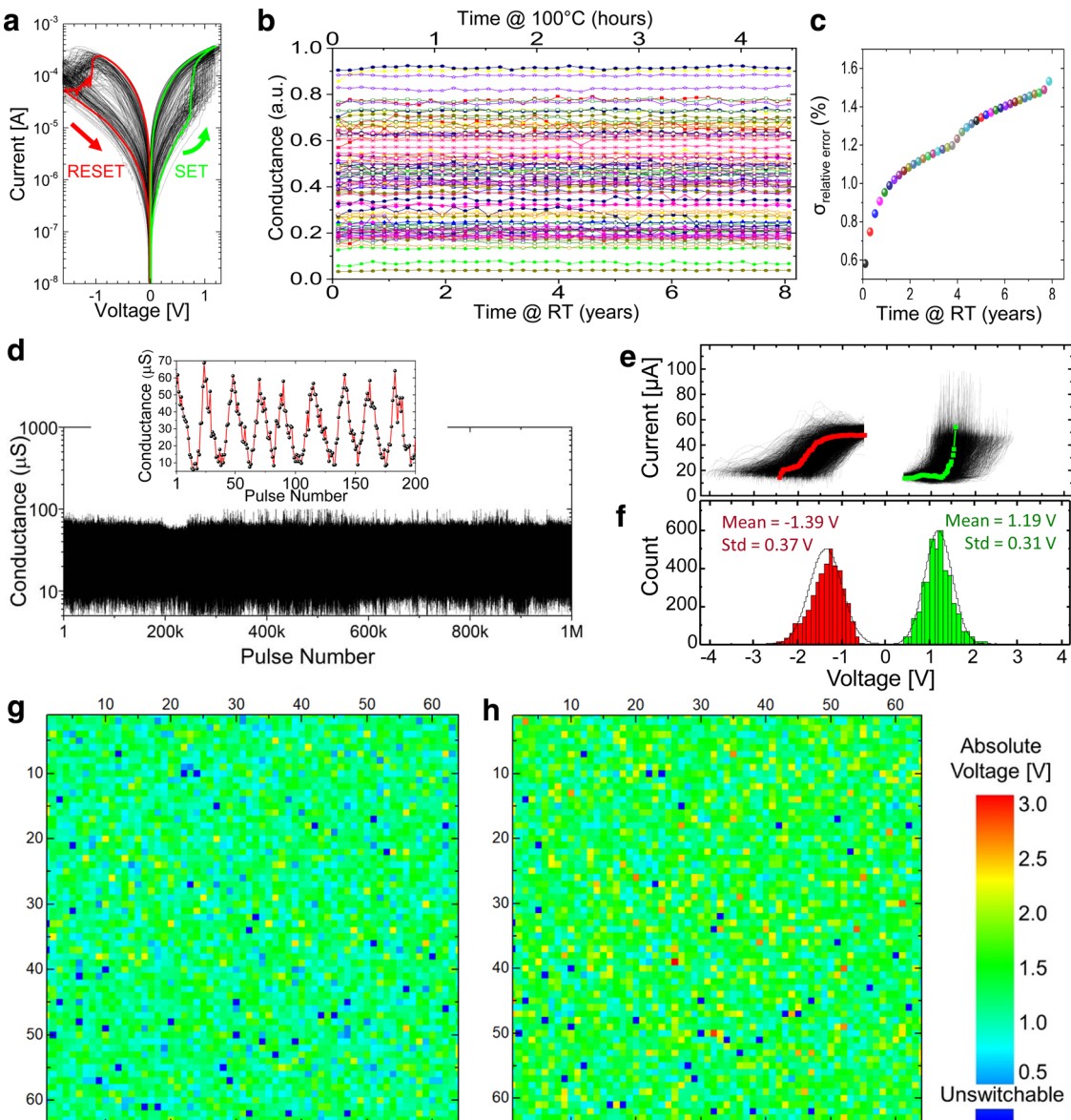

**Fig. 2 Device characterization results. a** Representative *I–V* curves, measured with quasi-static DC voltage sweeps, for the 36 formed devices of the 6 × 6 subarrays located in the center of the crossbar. For clarity, the curve for one particular device is highlighted. **b** Retention results for 10 different devices with data for each device shown with a specific color. The tests for each device are performed 9 times with randomly chosen initial conductance. The evolution of the conductance was measured at 400 s intervals while continuously baking the crossbar circuit at 100 °C. **c** The standard deviation of the absolute conductance change normalized to $G_{max} = 62.5\ \mu S$, i.e., $100\% \times |G_{initial} - G_{final}|/G_{max}$, as a function of the time interval for several ranges of initial conductances. Similar to panel **b**, the top axis corresponds to the measured retention data at 100 °C for 500 devices, with each device tested at 6 different initial states, while the bottom axis shows extrapolated results. For panels b and c, the bottom axes show extrapolated time at room temperature (RT) assuming 1.1 eV activation energy (see "Methods" section for details). **d** The switching endurance results for a crossbar device. The data are obtained by repeatedly applying alternative polarity sequences of 1-ms voltage pulses. The absolute amplitude of pulse in each sequence is initially 0.8 V and then ramped up with 0.1 V steps until the device reaches the extreme (i.e., on or off) state. Inset is a zoomed-in portion of the main panel, showing typical continuous switching during the endurance test. The device is switched about $10^5$ times between its extreme states during the experiment. **e** Measured evolution of conductance upon application of increasing amplitude voltage pulses. All parameters of the utilized pulse sequences are similar to those shown in Fig. 3c inset, except for the 50 mV incremental step. **f–h** Extracted statistics of switching threshold voltages, defined as the smallest absolute voltage at which device conductance, measured at 0.25 V, change by 20%, shown as (**f**) histogram and (**g, h**) voltage maps for (**g**) set and (**h**) reset transitions. The conductances are measured at 0.1 V for panels **b–d**.

memristors[17,24]. To electroform devices, a positive voltage is applied to the top electrode, while all unselected lines in the crossbar are floated[17]. Because of more extensive annealing compared to previous work[24], the currents via as-fabricated devices (Supplementary Fig. 4b) are just slightly less compared to the device's smallest (off-state) current after forming (Fig. 2a)— see, e.g., the highlighted curves for a specific device in both

figures. The forming voltages are only slightly higher on average than set voltages and completely overlap for some, making such devices effectively forming-free. The formed devices show similar magnitude set and reset voltages (Fig. 2a and Supplementary Fig. 5b), from 200 µA to 400 µA reset and set switching currents, 2-µA-to-50-µA dynamic current range at 0.25 V, and balanced *I–V* characteristics, i.e., $I(V) \approx -I(-V)$ at small voltages. The

average nonlinearities, i.e., $0.5 \times I(V)/I(V/2)$, are ~1.1 and ~1.3 for the on and off state, respectively, at $V = 0.25$ V.

Accelerated retention tests at 100 °C are performed for 500 devices, with each device tested in 9 different random states (Fig. 2b). Post-processed experimental results and their extrapolation for room temperature operation show very promising retention characteristics (Fig. 2b, c). For example, the extrapolated results predict that the normalized conductance will drift on average by ~0.7% over one month at room temperature. In comparison, the average spread is expected to be less than 1.6% after two years (Fig. 1c).

Figure 2d confirms excellent switching endurance. It shows the results of applying 1 million tuning pulses, or effectively, switching gradually device ~$10^5$ times between its extreme on and off states. Note that the experiment was stopped after reaching 1 M pulses because of the limitations of the experimental setup and not due to device failure. Furthermore, decent retention was observed even after the switching endurance experiment (Supplementary Fig. 5c).

Figure 2e shows measured switching dynamics characteristics for all the devices in the $64 \times 64$ array. These data are obtained by first setting the conductance of each device to 14 μS with 10% precision. Next, 1-ms-long pulses, with amplitude increased incrementally in 50 mV steps, are applied to the device. The device's conductance is read between each programming pulse at 0.25 V, and the sequence of pulses is stopped once the small-voltage conductance exceeded 50 μS. After that, we apply a similar reset/read pulse sequence until the conductance is switched back to 14 μS. The raw experimental data are used to extract effective switching thresholds, defined as the smallest amplitude of a voltage pulse at which the device conductance changes by more than 20% compared to its initial state (Fig. 2f–h and Supplementary Fig. 5a,b). According to Fig. 2f, the average set and reset threshold voltages are 1.19 V and –1.39 V, respectively, with the standard deviations of 0.31 V and 0.37 V. Furthermore, there are only 45 (~1.125%) unswitchable devices in the whole crossbar array. The threshold maps show that faulty devices are distributed throughout the array and not contributed by faulty lines but rather stand-alone defects. These failed devices are most likely due to applying insufficiently high forming/switching voltages, which we had to bound as a precaution for avoiding permanent damage to the crossbar circuit. This, in part, is supported by the tails of the distribution in the switching threshold voltages. Interestingly, there is a weak positive linear correlation between set and reset voltage amplitudes (Supplementary Fig. 5b). Additional experimental data on switching dynamics are collected for model development—see Supplementary Fig. 7a–d and its the discussion below.

**Conductance tuning experiments**. The analog properties of the memristive crossbar circuits are tested by setting crosspoint device conductances using the fine-tuning algorithm[39]. Such an algorithm, similar to incremental step programming of flash memory devices, is based on applying a sequence of smaller-voltage non-disturbing read and larger-voltage write pulses, with a sign and amplitude of write pulses are adjusted dynamically based on the measured conductance at read pulses. An example of applying such a write-verify algorithm is illustrated in Fig. 3a, which shows the evolution of the low-voltage conductance of a specific device upon its forming, resetting to 20 μS and then tuning to 10 μS, 100 μS, and 8 μS target conductance values. Note the polarity of the tuning pulses in the inset—while applying both sets and reset pulses were required because of the overshooting for tuning to 8 μS and 20 μS, only gradual resetting (setting) was sufficient to tune to 10 μS (100 μS). In fact, the device

conductances can be precisely set to any value in a range from ~2 μS to ~100 μS—see, e.g., the results of device tuning with 1% relative error to linearly spaced conductance values within the lower half of the dynamic range in Fig. 3b. Figure 3c–e shows the results of tuning conductances of all devices in the crossbar circuit, using write pulses with up to 2.5 V maximum amplitude and 4 mV/8 mV incremental step for set/reset (Fig. 3c, inset). To reduce disturbance of already tuned half-selected devices in passively integrated crossbar circuits, the half-biasing scheme is adopted when applying write pulses[17] (Supplementary Fig. 1). Furthermore, to correct for a minor conductance drift in some half-selected devices upon programming, tuning of the whole crossbar is performed in several rounds, such that, e.g., all of the devices are tuned, one by one, in the first round, and then those which got disturbed beyond the specified tuning accuracy are re-tuned in the following round(s). In particular, Fig. 3d shows the map of target conductances, representing the gray-scale image of Albert Einstein mapped on all devices in the $64 \times 64$ crossbar array, while Fig. 3e shows their final values after three rounds of tuning. The corresponding statistics for the relative tuning error are shown in Fig. 3c. Excluding unswitchable devices, for which the error is more than 95%, ~98% of the devices are tuned within 5% relative error, while the average relative error is ~3.76%.

In the previous experiment, the tuning algorithm is stopped once the desired 5% relative tuning error is reached. Setting conductances with even higher precision is already demonstrated for tuning a specific device in the crossbar in Fig. 3b. The possibility of achieving higher tuning precision at the circuit level is indirectly indicated by the shape of the tuning error histogram in Fig. 3c and further verified by implementing an ex-situ trained image classifier and testing it on the common MNIST hand-written digit benchmark[40] (Fig. 4). In this experiment, we focus on demonstrating vector-by-matrix multiplication, the core operation in any neural network, while the functionality of neurons, including its bias, is emulated in the software. For simplicity, the studied network is a single-layer perceptron with 64 inputs, 10 outputs, and 640 weights. Furthermore, the original binary $28 \times 28$ MNIST images are down-sampled to $8 \times 8$ patterns, so that they can be represented with 64-bit binary vectors in which black/white pixels are encoded by 0 V/0.25 voltages and are applied to the vertical crossbar lines. Each weight is implemented with one memristor using a 10 μS to 110 μS range of conductances by shifting the range of the weights upon mapping and adding pattern-dependent neuron bias at the post-processing stage—see "Methods" section for more details. By encoding network weights with the corresponding memristor conductances $G$ in the $64 \times 10$ portions of the crossbar, the currents measured at the virtually grounded horizontal lines of the crossbar represent the results of vector-by-matrix multiplication operation, while the output with the largest current identifies the computed class of the input pattern (Fig. 4a).

The measured classifier fidelity and the software-based performance of the same network across a 1% to 50% range of weight import errors are shown in Fig. 4d. The results show that the experimental data match simulation results closely. For example, the measured classifier accuracy for the most accurate weight import is 1.87% lower than that of the ideal software model, while the average and standard deviation for the neuron pre-activation errors are 0.61% and 0.37%—see additional details in Supplementary Fig. 6b, c. Note that the goal of this experiment is to demonstrate the conductance tuning capabilities rather than on demonstrating high classification accuracy, which is quite low compared to the state-of-the-art numbers because of the utilized single-layer network and down-sampled B/W images. However, it is worth mentioning that the high accuracy MNIST benchmark results for the mixed-signal circuit RRAM-based implementations

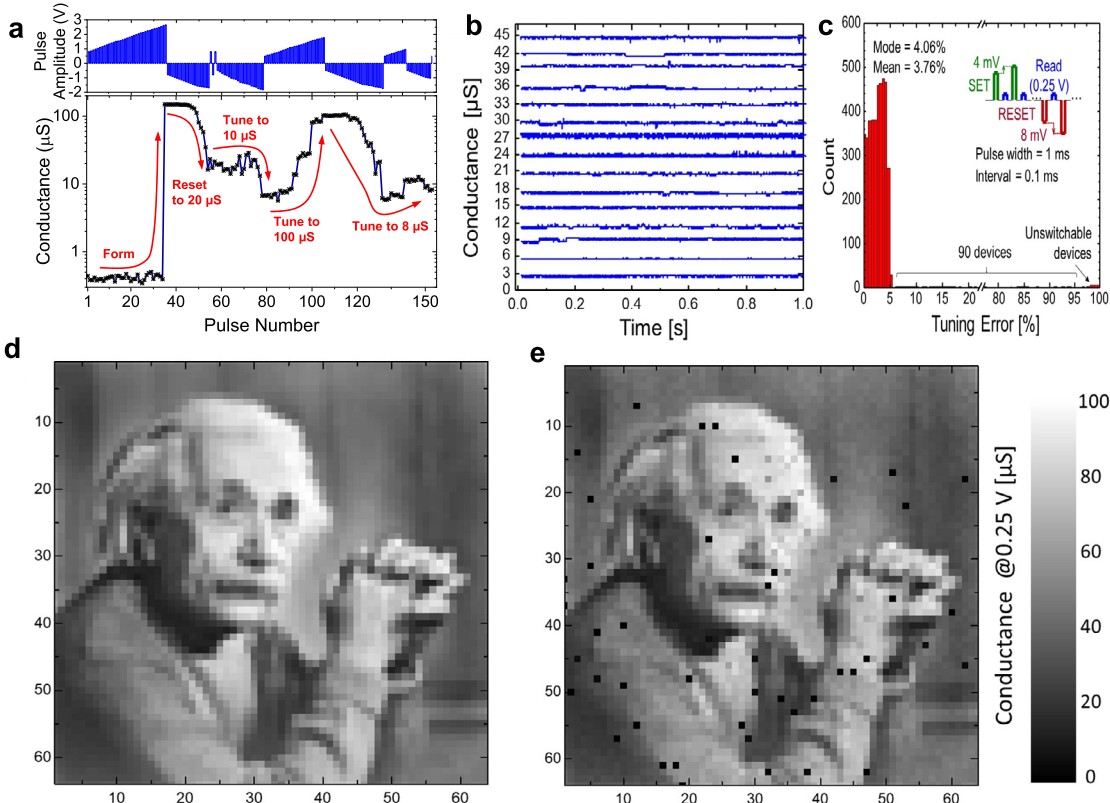

**Fig. 3 Conductance tuning results. a** Forming and high-precision tuning to 20 μS, 10 μS, 100 μS, and 8 μS target conductances of a crossbar device, with 1% relative precision. The inset shows the applied sequence of pulses during conductance tuning. Pulse sequences parameters are similar to those of panel **c**, except for the utilized 50 mV incremental step. **b** Example of device tuning with a 1% relative error to different conductance levels equally spaced from 3 μS to 45 μS. **c–e** Programming Einstein image to the 64 × 64 crossbar array with a 5% relative error. **c** Tuning statistics. Inset shows details of the write-verify pulse sequence. **d** The target device conductances in the range of 10 μS to 100 μS corresponding to the gray-scale quantized image and (**e**) their actual measured values after completing automated tuning. The relative tuning error is defined as $100\% \times |I_{target}(0.25\,V) - I_{actual}(0.25\,V)| / I_{target}(0.25\,V)$. All conductances are specified at 0.25 V. Einstein image copyright by Yousuf Karsh.

are sometimes obtained via hybrid demonstration. For example, the reported high classification accuracy in ref. [28] was due to using a very high complexity network in the simulations, while the experimental results for the 0T1R circuit were only performed for 3 × 3 filters—see, e.g., largest working demo complexity column in the Supplementary Table 1.

Figure 4e, f provides more details on the measured data for the two representative MNIST patterns. Specifically, the first examples show the results of the correct classification of pattern "7", with the largest current measured at the 7th row of the crossbar (Fig. 4e). On the other hand, pattern "9" in the second example is misclassified (Fig. 4f). This is in part because of a large tuning error at unswitchable memristors—see stuck at high-resistance state devices at (9, 22) and (9, 24) locations in the crossbar in Fig. 4c (and also Fig. 2g, h). It is also due to narrow current margins between the correct class and the two closest classes representing digits "0" and "8", which is natural given that correct classification, in this case, would be challenging even for a human.

**Modeling of advanced systems.** We next investigate the prospects for tuning algorithm improvements and algorithm's application in ex-situ-trained neuromorphic inference accelerators. To make this study more informative, we develop the model for the conductance tuning process and then investigate the impact of device variations on the circuit functionality. A

specific focus is on the modeling of half-select disturbance, which is a major challenge for accurate conductance tuning, as confirmed by experimental work. Similar to the previous work[41], dynamic phenomenological model capturing device-to-device variations is derived by fitting experimentally observed conductance changes for 500 crossbar integrated memristors upon application of write voltage pulses with variable amplitude—see Supplementary Note 1 and Supplementary Fig. 7 for more details of the model.

Using the developed model, classifier accuracy is simulated for the ex-situ-trained 784-64-1 multilayer perceptron network implemented with a hybrid CMOS/memristor circuit under various assumptions of device-to-device switching threshold variations. Specifically, using the differential pair encoding of the weights (Supplementary Fig. 8a), the 785 × 64 weight layer, with the additional input due to bias, is mapped to 24 64 × 64 and 2 17 × 64 mixed-signal VMM blocks. In addition to the memristive crossbar array, each block hosts a digital-to-analog converter (DAC), local sensing based on transimpedance amplifier, and programming circuitry (Fig. 5a). Such distributed implementation is similar to a mixed-signal architecture of the aCortex[42,43], in that the output of the local sensing circuits are currents corresponding to the partial dot-products between the corresponding weights and inputs, while the full dot-products are computed by the neuron's (global sensing) transimpedance amplifiers by summing partial product currents. The hidden layer neurons then compute clipped rectified linear function activation and pass the results to the second layer of the

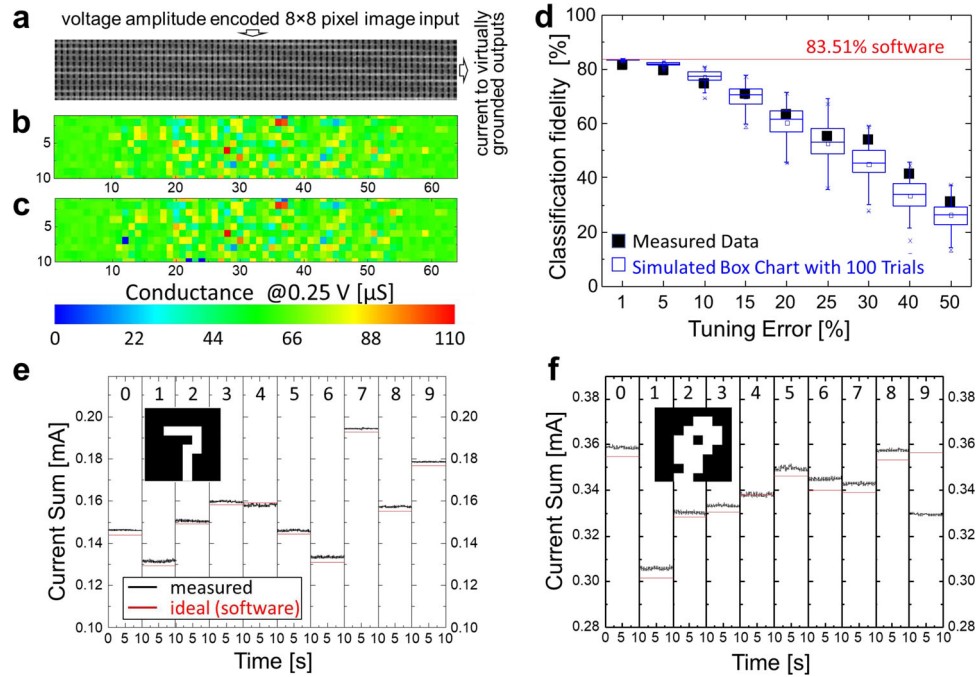

**Fig. 4 Experimental results for pattern classification. a** A portion of the crossbar circuit is utilized in a 64 × 10 single-layer perceptron MNIST image classification experiment. **b** Examples of target and (**c**) actual conductances after tuning with a 1% relative error. **d** Measured classification fidelity and its comparison with simulation results as a function of weight import accuracy. In each simulation trial, the weights are selected randomly from a range of target_value × [1 − tuning_error, 1 + tuning_error]. **e, f** Measured output currents for all ten outputs over the 10-s interval for patterns '7' and '9' (shown in the corresponding insets) for the experiment with a 1% relative tuning error. The currents are measured, one row at a time, by simultaneously applying input voltages on all 64 columns and grounding 10 specified rows. See Fig. 3 caption for the definition of relative tuning error.

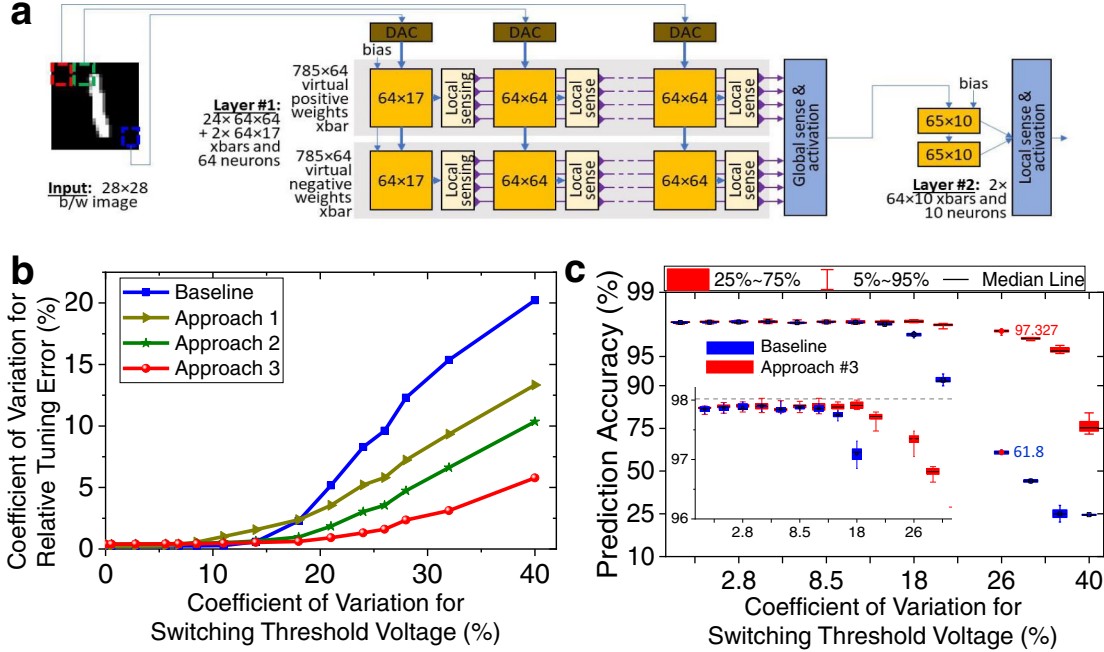

**Fig. 5 Modeling of ex-situ trained MLP classifier. a** The block diagram for the distributed mixed-signal implementation of 784-64-10 multilayer perceptron (MLP) classifier with 64 × 64 crossbar circuits. Programming circuitry is omitted for clarity. **b** Modeled relative tuning error after 10 rounds of tuning for the first layer of MLP network as a function of device-to-device variations when using four different conductance tuning approaches. See Section "Modeling of Advanced Systems" for more details of the tuning approaches. **c** Simulated accuracy of MLP classifier as a function of device-to-device variations when using baseline and the most advanced tuning approach after 10 rounds of tuning. Inset shows zoom-in for the high classification accuracy portion of the graph. The shown numbers roughly correspond to the device variations observed in the experiment. The box plot shows the statistics over 10 different runs of initial conductances. For simplicity, memristors' static I–V nonlinearities and noise are neglected, and ideal peripheral circuits are assumed in simulations. See Fig. 3 caption for the definition of relative tuning error.

network. A similar, though simpler due to the analog nature of input signals, implementation is assumed for the second layer of the network, consisting of two $64 \times 10$ analog VMM circuits. (Note that the VMM block dimensions are chosen to match experimental work and not necessarily optimal for the studied parameters of memory devices.) More details on the network training and modeling assumptions are provided in the "Methods" section.

In the first studied "baseline" approach, the devices are tuned in the sequential (raster) order, similar to the experimental work. The results for the baseline algorithm show that both the tuning and classifier accuracies are significantly degraded due to half-select disturbance when the device-to-device variations (i.e., coefficient of variation in switching threshold) are above 14% (Fig. 5b, c, and Supplementary Fig. 9a–c). The second round of tuning increases the accuracy significantly, though the improvements with additional rounds are negligible (Supplementary Fig. 9a). The simulated absolute tuning error at 26% device variations is ~9.6% (Fig. 5b), which is higher than the experimental results because of the included (unswitchable) devices.

Three different techniques are further proposed to improve the conductance tuning process. In the first technique, the write voltage amplitudes are bounded within a certain range of voltages, with the range gradually reduced with each round of tuning (see Supplementary Fig. 9). Such an approach results in better average tuning accuracy than the baseline approach when device variations are higher, at the cost of abandoning tuning the devices with larger threshold switching voltages (Fig. 5b). In the second technique, the devices with a high (>1.75 V) set and high (<−2 V) reset switching voltages are first identified. The high-set threshold devices are then switched to the highest conductive (>75 μS) state used in weight mapping, while the high-reset devices are switching to the highest resistive (<7.5 μS) before the tuning algorithm is applied. Such presetting significantly reduces the use of larger amplitude write pulses throughout the tuning process, and hence minimize half-select disturbance, especially when applied together with the first technique—see the results for approach #2, which utilizes both techniques in Fig. 5b. The third technique takes advantage of the possibility to encode the same weight with different target conductances in the differential pair implementation, i.e., by shifting the conductances of a pair by the same amount. In particular, when the maximum voltage limitation of the first technique is met, the target conductances of a pair are adjusted, and the conductance tuning of another device in a pair is attempted instead. Application of all three techniques (approach #3) significantly improves the tuning accuracy, e.g., improving it by 9% compared to the baseline approach for the case of 26% device variations. More importantly, at such device variations, the classification accuracy of the baseline approach is significantly improved to ~97.3%, which is within 0.7% of the highest possible accuracy for the studied network, while the highest amount of device variations, which can be tolerated without losing classification accuracy is increased from ~14% to ~20% (Fig. 5c and Supplementary Fig. 9c, f, h, l).

## Discussion
The high prospective integration density of passive memristive crossbar circuits, enabled by both aggressive lateral feature scaling and vertical monolithic integration, would be essential for hardware implementations of large neural network models, such as those used for the end-to-end automatic speech recognition, natural language translation, and text summarization, on a single chip without having to perform very energy-taxing and slow data transfer with the off-chip memory. For example, the largest multilingual neural model for automatic translation among seven

common languages contains 640 million parameters[44]. The functional performance of the transformer networks, the state-of-the-art models for text summarization, dramatically improves with the network scale, e.g., almost linearly improving when increasing the number of parameters in GPT-2 model from few hundreds to ten billion[45]. Furthermore, mixture-of-expert networks with up to 137 billion parameters have been recently suggested to improve the functional performance of language modeling[46]. Storing that many parameters on-chip could be hardly accommodated with planar embedded memory technologies. Though the complexity of the mentioned above large-scale neural networks might reduce with further improvements in algorithms, it is clear that extremely large models will still be useful. This can be indirectly evidenced by the complexity of the human brain, which, with its ~$10^{15}$ synapses, can serve as a proxy for the complexity of the future highly cognitive neuromorphic systems[47].

The importance of memory density is indirectly confirmed by earlier work on general-purpose neuromorphic inference "aCortex" accelerator based on embedded NOR-flash technology. The modeling results showed that memory devices could occupy up to 25% of the total area, while the remaining area was devoted to peripheral circuits and other functions[42], even though aCortex utilizes moderate-size $64 \times 64$ VMM circuits. (aCortex implementation with larger VMMs circuits was less area-efficient because of the higher amount of underutilized crossbar devices when mapping common neural network models.) Our crude estimates show that even with largely suboptimal technology and moderate-size $64 \times 64$ VMM circuits, aCortex and fully-analog MLP circuit implementations based on passively integrated memristors have almost two times smaller areas compared to those of 1T1R technology (Supplementary Note 2 and Tables 2–4). The memory efficiency (i.e., the memory density importance) and the performance gap between 1T1R and 0T1R based implementations become larger when memory cell currents and switching/forming voltages are decreased (Supplementary Fig. 11) and/or when implementing more-specialized circuits, such as large-scale models that do not rely on weight sharing and could benefit from larger crossbar array implementations. Making larger crossbar circuits would require additional technology advances, most notably increasing the ratio of an electrode to device conductance and improving the device uniformity. For example, the former can be achieved by reducing leakages within the device and between neighboring lines, e.g., by patterning the active switching layer and scaling down device feature sizes[24], and making higher-aspect ratio electrodes, e.g., similar to those utilized in DRAM memories.

While this paper is focused explicitly on analog-grade (i.e., multi-bit) devices, low-precision (e.g., binary weight and/or binary activation) neural network models have also received significant attention[19,20,25,28,48]. However, it seems that understanding and dealing with the impact of reduced weight and computing precisions is still a very active area of research. For example, though little or no loss in accuracy can be achieved when using binary weights for some of the earlier (very redundant) deep convolutional networks, such as AlexNet or VGG, 4 to 8 bits of precision for both weights and activations might be necessary for the most advanced image classifiers[49]. A related observation is that the accuracy loss can often be recovered by increasing the network depth and/or width[50,51], which, however, naturally results in decreased physical performance. Higher precision weight can also be implemented using multiple lower-precision memory devices[52]. In this case, multiple VMM circuits are employed for different significance portions of the weight values. VMM operation is performed by first calculating partial VMM outputs and then properly scaling and adding such outputs

with the peripheral circuitry to obtain the final result. Ultimately, the prospects for lowering precision in the neural network or employing redundant designs, which might enable using simpler binary ReRAM devices, can only be understood by considering both functional and physical performances at the system level[53].

In summary, the general goal of this work is on increasing the complexity of passively integrated memristive crossbars and developing a fully CMOS-compatible process while maintaining high yield and sufficiently low spread in current-voltage characteristics of integrated metal-oxide memristors, one of the most critical problems prohibiting practical use of this technology in neuromorphic computing applications. Our main contributions include the development of uniform $64 \times 64$ passive crossbar circuits with almost 99% working crosspoint metal-oxide memristors based on foundry-compatible fabrication process suitable for back-end-of-line/3D integration and experimental demonstrations of conductance tuning with <4% relative average error for programming 4 K gray-scale pattern and close to 1% error when implementing 640-weight ex-situ-trained single perceptron network. In addition, we propose the advanced tuning algorithm and verify its effectiveness by simulating a multilayer perceptron. We believe that our results are a significant improvement in both complexity and analog properties over previously reported passive crossbar memories and an important step towards realizing human-brain-scale integrated neuromorphic systems. The near-term work should focus on improving technology to increase yield and reduce device variations, decrease write and operating currents of memristors, and ultimately demonstrate practical fully integrated hybrid circuits, e.g., with back-end-of-line fabricated memristors on top of the CMOS subsystem that would outperform purely-CMOS counterparts. Furthermore, theoretical efforts should focus on developing holistic circuit and algorithm techniques for coping with device variations and faulty devices.

## Methods

**Crossbar fabrication**. The first step in the fabrication is the deposition of Ti (10 nm)/Al (70 nm)/TiN (45 nm) metal stack on a 4-inch Si wafer with 200 nm of thermally grown $SiO_2$ using reactive sputtering (Fig. 1a, b). ~250-nm wide bottom electrodes are then patterned by deep ultraviolet lithography stepper with an antireflective coating (Brewer Science DUV-42P) using a negative photoresist (Dow Chemical UV2300-0.5) and inductively coupled plasma etching process with $BCl_2/Cl_2/N_2$ chemistry to suppress sidewall re-deposition during etching (Fig. 1c). The bottom electrodes are planarized by first depositing 300 nm of $SiO_2$ via plasma-enhanced chemical vapor deposition (Fig. 1d). The chemical-mechanical polishing (CMP) process is then used to smoothen the $SiO_2$ surface, which is followed by etch-back with $CHF_3$ plasma (Fig. 1h) to open bottom electrodes. The thicknesses of the remaining $SiO_2$ after CMP are measured by ellipsometer (unlike calibration via slow etching and atomic force microscope imaging used in ref. [22]). The $Al_2O_3$ (1.5 nm) and $TiO_{2-x}$ (30 nm) of the active switching bilayer are deposited, respectively, through atomic layer deposition and reactive sputtering (Fig. 1g). No oxygen descum is conducted after switching layer deposition to keep $TiO_{2-x}$ stoichiometry. Approximately 250-nm top electrode lines with Ti (15 nm)/Al (90 nm)/TiN (80 nm) are deposited and patterned similarly to the bottom electrodes (Fig. 1e, f). The switching layer outside the crossbar region is etched with $CHF_3$ plasma to suppress line-to-line leakages and open ends of bottom electrodes. Ti (40 nm)/Au (400 nm) pads are formed for wire bonding and packaging. Finally, rapid thermal annealing at 350 °C in $N_2$ gas with 2% $H_2$ for 1 min is performed after the crossbar fabrication is complete.

**Electrical characterization**. The crossbar array circuit is wire-bonded and mounted on a custom printer board for testing and application demonstration. The custom-printed circuit board is connected to Keysight tools and controlled by computer setup (Supplementary Fig. 10). All electrical measurements are performed using the Keysight B1500A parameter analyzer. The connections to crossbar inputs/outputs are steered by the Keysight 34980 A switching matrix. The parameter analyzer and the switching matrix are connected to a personal computer via a general-purpose interface and universal serial buses and controlled using a custom C++ code.

**Retention extrapolation**. The rate of conductance change is approximated using Arrhenius law, i.e., rate $\propto \exp[-U_A/(k_B T)]$, where $U_A$ is an activation energy for the memory mechanism, $k_B$ is a Boltzmann constant, and $T$ is ambient temperature. Using this equation, the predicted time interval $t_{293K}$ for the conductance change $\Delta G$ at room temperature (293 K) is expressed via the observed time interval $t_{373.15K}$ over which the conductance was changed by the same amount at elevated 100 °C ($\equiv$373.15 K) temperature, i.e., $t_{293K} \approx t_{373.15K} \exp[U_A/k_B \times (1/293[K] - 1/373.15[K])]$. Such an extrapolation approach is in line with previous theoretical and experimental studies of metal-oxide memristors, which implies retention loss through temperature-activated drift of oxygen vacancies[54,55]. The extrapolation results in Fig. 2b, c are shown for $U_A = 1.1$ eV, corresponding to the oxygen vacancy activation energies in rutile phase titanium dioxide[56].

**Image classifier details**. The experimental work in Fig. 4 is based on the $64 \times 10$ single-layer perceptron classifier. The network is trained in software on a 60,000 training and 10,000 $8 \times 8$ down-sampled MNIST test images using the conventional back-propagation algorithm with 0.01 learning rate, 100 batch size, and 50% dropout rate to find the weights ($w$) and biases ($b$). The weights are then linearly mapped to the conductances of corresponding crossbar devices via the equation $G = c_1 \times w + c_2$, where $c_1 = 10^{-4}/\text{Max}[w - \text{Min}[w]]$ and $c_2 = 10^{-5}$–$10^{-4} \times \text{Min}[w]/\text{Max}[w - \text{Min}[w]]$ are scaling constant and weight bias, respectively, which are selected to map arbitrary dimensionless weight range to the range of conductance from 10 µS to 110 µS. Note that the shift in the mapping requires subtracting a pattern-dependant term $c_2\sum x_i$ when calculating neuron outputs. Specifically, with such implementation $\sum w_i x_i + b \propto \sum G_i v_i + I_B$, where $\sum G_i v_i$ is measured experimentally, as discussed in the main text, while $I_B = 0.25c_1 b - c_2\sum v_i$ is an additional current bias added to the neuron at the post-processing stage, where the factor of 0.25 due to mapping of inputs $x = 1$ in the software to the applied voltages $v = 0.25$ V in the experiment.

The modeling work in Fig. 5 is based on the 784-64-10 multilayer perceptron classifier with rectify-linear hidden layer neurons. The inputs to the first layer, i.e., pixel intensities, are linearly mapped to [0 V, 0.1 V] voltage range, while the inputs to the second layer are also in the same range, due to the assumed clipping of rectified linear function at the neuron side. The classifier is trained ex-situ on a gray-scale 60,000 training and 10,000 MNIST test images using the conventional backpropagation algorithm with L2 regularization, 0.0005 learning rate, and 100 batch size. The software weights ($w$) are converted to the corresponding pair of positive ($G^+$) and negative ($G^-$) conductances using differential mapping with 33.75 µS range and 41.25 µS bias, which is roughly in the middle of the dynamic range, i.e., $G^\pm = 41.25$ µS $+ 16.875$ µS $\times w/w_{\text{max}}$, where $w_{\text{max}}$ is specific to the MLP layer largest absolute weight value (Supplementary Fig. 8b–f). The weight import process is simulated by first randomly initializing conductances of all memristors according to the normal distribution with a 36.25 µS average and 9 µS standard deviations. A tuning algorithm based on 5-mV-step increasing amplitude pulses, starting from 0.5 V, is then applied with a 1% desired tuning accuracy, which is sufficient for achieving the highest classification accuracy with no device-to-device variations. To limit each device's tuning time, the number of times for switching the write pulse polarity (when overshooting the target conductance) is limited to 5.

## Data availability
The data that support the plots within this paper and are available from the corresponding author upon reasonable request.

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

## Acknowledgements

This work was supported in part by a Semiconductor Research Corporation (SRC) funded JUMP CRISP center, and in part by NSF/SRC E2CDA grant 1740352. The authors would like to thank Dr. Mirko Prezioso and Dr. Ilya Karpov for useful discussions.

## Author contributions

H.K. and H.N. fabricated the crossbar. H.K. and M.R.M. performed the experiments. M.R.M and H.N. built the experimental setup. M.R.M. developed the model and performed the simulations. D.B.S. wrote the manuscript. All authors discussed the results. All authors contributed equally to this work.

## Competing interests

The authors declare no competing interests.
