## [Peer Review File · Nature Communications]

Reviewers' Comments:

Reviewer #1:

Remarks to the Author:

The responses have well addressed the reviewer's previous comments. There are a couple additional questions and suggestions.

1. In Supplementary Table 1, could the authors add information on how many levels of conductance each work's analog RRAM design demonstrated?
For the 1T1R, could the authors add the work of <https://ieeexplore.ieee.org/document/9145716> in the comparison?

2. Regarding 1T1R design, while the authors claim that they are focusing on analog-grade, a way to achieve 16 levels of conductance from binary RRAM devices is to use multiple RRAM cells. For example, ~four binary RRAM cells could be used for 16 levels of conductance as in <https://www.nature.com/articles/s41928-020-00505-5>, or two 4-level RRAM cells could be used (if multi-level industrial RRAMs as in <https://ieeexplore.ieee.org/document/9145716> are employed). These approaches will be more area-efficient than the $375F^2$ analog 1T1R cell reported in Supplementary Tables 3 and 4.

In Supplementary Tables 3 and 4, could the authors add one more column on the right for the scheme of using multiple digital 1T1R cells? That would be a good comparison to a more commonly understand "baseline".

Reviewer #2:

Remarks to the Author:

The authors have taken much effort in addressing some of my concerns. But I cannot agree with the point that "advantage of 1T1R neuromorphic circuit over the conventional digital circuits". Such advantage in energy efficiency has been demonstrated in the application of convolutional neural network (e.g. *Nature* 577, 641 (2020)). Although the passive memristor crossbar arrays may have potential in higher integration density than 1T1R based, its practical application either in memory or neuromorphic computing could be greatly affected by the unique operation, i.e. half-select disturbance, which was agreed by the authors. As an attempt in fabricating larger passive memristor array than ever before, this work represents an important step forward possible application of 0T1R array. Based on this, I would not further insist on other points on the advantage of 0T1R array over 1T1R.

Reviewer #3:

Remarks to the Author:

The authors have addressed most comments that were raised in the last round and added insightful comments on the feasibility and necessity of making larger transistor-less crossbars. The reviewer believes the manuscript is improved and would like to support its publication, but it looks like there is one comment from the last that remain unaddressed:

"1. The discussed array conductance tuning algorithm is very interesting, but it is not very clear to the reviewer whether the authors have implemented the best tuning scheme (approach 3) in their experiments. If not yet why not?"

Response to the Reviewers' Comments

We are once again grateful for reviewers' comments. Please see our inline responses below

Reviewer #1 (Remarks to the Author):

The responses have well addressed the reviewer's previous comments.

Thank you for helping us to improve the paper.

There are a couple additional questions and suggestions.

1. In Supplementary Table 1, could the authors add information on how many levels of conductance each work's analog RRAM design demonstrated?

Thank you for this comment. We believe that this information is already in the table, i.e. can be uniquely determined from the array-level tuning precision, which was typically reported across the whole dynamic range, and G_{max}/G_{min} range information.

For the 1T1R, could the authors add the work of <https://ieeexplore.ieee.org/document/9145716> in the comparison?

We were not aware of this JSSL (<https://ieeexplore.ieee.org/document/9145716>) paper. Thank you for bringing it to our attention! We believe it is a great work that shows the advances in the field and will help progression of the neuromorphic ReRAM circuits. We are now citing it in the revised paper in the context of binary activation networks, i.e.

"While this paper is focused explicitly on analog-grade (i.e., multi-bit) devices, low-precision (e.g., binary weight **and/or binary activation**) neural network models have also received significant attention [19, 20, 25, 28, 55]."

We wish, however, that more crucial details in that paper were provided. In JSSL and in the original paper on the utilized ReRAM technology (Ref. 10 in JSSL), we could not find any information on the write/forming currents. The units in the I-V plot (Fig. 5 of Ref. 10) are omitted, while JSSL paper discusses operation in terms of G_{high}/G_{low} or shows figures with a.u. rather than absolute values.

Such information is extremely important because write/forming currents and voltages statistics defines the practical values for the sizing of access transistor and hence the area of 1T1R cells. It could be that JSSL and Ref. 10 from JSSL report technology with very small 1T1R cell. Alternatively, it could be due to very aggressive scaling of access transistor in JSSL design which could have resulted in inability to form or switch significant fraction of ReRAM devices (which are at the upper end of the tail for write voltages/currents distribution). Fig. 3 in JSSL is impressive but it is not clear if it is reported for a subset of cells or for the whole array, e.g. for 4K ReRAM cells as it is mentioned in text when it seems that there are 8K total cells in the fabricated array. Also confusing is why Ref. 10, which is cited extensively in JSSL as the utilized ReRAM device technology, reports $56F^2$ area per 1T1R cell, while JSSL reports $31F^2$. It is quite probable that lack of these details is not a fault of authors but rather have been enforced by company which provided ReRAM technology. However, even if it is so, we hope that reviewer would agree that the clarity in these details is extremely important for making fair comparisons.

2. Regarding 1T1R design, while the authors claim that they are focusing on analog-grade, a way to achieve 16 levels of conductance from binary RRAM devices is to use multiple RRAM cells. For example, ~four binary RRAM cells could be used for 16 levels of conductance as in

<https://www.nature.com/articles/s41928-020-00505-5>, or two 4-level RRAM cells could be used (if multi-level industrial RRAMs as in <https://ieeexplore.ieee.org/document/9145716> are employed).

We fully agree with this point and added the following text in the paper:

“Higher precision weight can be also implemented using multiple lower-precision memory devices [56]. In this case, multiple VMM circuits are employed for different significance portions of the weight values. VMM operation is performed by first calculating partial VMM outputs and then properly scaling and adding such outputs with the peripheral circuitry to obtain the final result. Ultimately, the prospects for lowering precision in the neural network or employing redundant designs, ...”

These approaches will be more area-efficient than the $375F^2$ analog 1T1R cell reported in Supplementary Tables 3 and 4. In Supplementary Tables 3 and 4, could the authors add one more column on the right for the scheme of using multiple digital 1T1R cells? That would be a good comparison to a more commonly understood “baseline”.

Please first note that our assumption for $375F^2$ is based on the actual design layout which utilizes high V_t transistors from 65-nm PDK to accommodate forming/switching currents and voltages that are relevant to our ReRAM devices. All of these details are outlined in Tables S3 and S4 and the SI text.

We would agree with the Reviewer that if we assume lower forming/switching currents (and/or some specific denser high V_t transistor processes) the area of the access transistor can be reduced. However, better area efficiency would come in this case not from better VMM design - in fact design with multiple devices per weight would have worse performance / energy efficiency - but rather more optimal 1T1R ReRAM cell technology. Furthermore, making more aggressive ReRAM / high V_t assumptions would also benefit 0T1R design in Table S3 and S4, e.g. reduce overhead for high- V_t local sensing and analog mux circuitry as it should be clear from Tables S3 and S4. So we believe that making a fair comparison would require changing the assumptions about 0T1R and 1T1R cells and hence recalculating all the columns in Table S3 and S4, not just extending it as Reviewer is asking. On the other hand, changing ReRAM assumptions would make it less relevant to our work – we would like to kindly remind the Reviewer that our paper is not about the VMM design but rather device technology.

Furthermore, to add the new results based on JSSL paper that reviewer is cited in his comment we would need to know exact values for forming / switching / read currents and voltages, which are unfortunately completely missing in the JSSL paper – please see our response above.

Please note that we already included a similar albeit simpler modeling study in Supplementary Figure 11 in which we modeled VMM array performance as a function of the forming/switching current values. For simplicity, it was performed at the VMM array level, because performing it at the system level as in SI Tables 4 (i.e. designing an optimal CMOS circuitry for each datapoint) in such an exploration study would not seem reasonable (please see our responses in a previous round of review for more details).

Finally, we believe that SI information is already too busy and has a lot of material which might be distracting from the main message of the paper, which is the demonstration of highly uniform largest passive crossbar with tunable conductance nonvolatile crosspoint device.

Reviewer #2 (Remarks to the Author):

The authors have taken much effort in addressing some of my concerns. But I cannot agree with the point that “advantage of 1T1R neuromorphic circuit over the conventional digital circuits”. Such

advantage in energy efficiency has been demonstrated in the application of convolutional neural network (e.g. Nature 577, 641 (2020)). Although the passive memristor crossbar arrays may have potential in higher integration density than 1T1R based, its practical application either in memory or neuromorphic computing could be greatly affected by the unique operation, i.e. half-select disturbance, which was agreed by the authors.

The reviewer would probably agree that PCB integrated system, e.g., with neurons and memory arrays implemented on different ICs, can hardly compete in performance with fully integrated CMOS ICs and the technology should be much more refined (better yield etc.) to enable truly integrated systems. We also believe that a fair, insightful comparison when evaluating emerging neuromorphic technologies can only be performed at the system level. For example, it could be a comparison of Joules per classification inference per image frame between two systems at similar implemented classification accuracy. Such comparison would take into account all peculiarities of the technologies (yield, computing precision etc.) and chosen models (redundancy etc.), as opposed to comparing Ops per joule as it was done in that Nature paper, which can be very misleading.

In any case, we strongly believe in ReRAM technology ourselves and believe that Nature 577, 641 (2020) is a state of the art work that is advancing the field. It seems that we just have a different opinion about this nontechnical issue that is not directly related to our work.

As an attempt in fabricating larger passive memristor array than ever before, this work represents an important step forward possible application of 0T1R array. Based on this, I would not further insist on other points on the advantage of 0T1R array over 1T1R.

We really appreciate this positive feedback!

Reviewer #3 (Remarks to the Author):

The authors have addressed most comments that were raised in the last round and added insightful comments on the feasibility and necessity of making larger transistor-less crossbars. The reviewer believes the manuscript is improved and would like to support its publication, but it looks like there is one comment from the last that remain unaddressed: ...

Thank you for this comment!

... "1. The discussed array conductance tuning algorithm is very interesting, but it is not very clear to the reviewer whether the authors have implemented the best tuning scheme (approach 3) in their experiments. If not yet why not?"

We apologize for not answering to that comment in the first round of revision. We have been very careful in not creating ambiguity in that issue and clearly and explicitly stated in the paper that all the results to better tuning approaches are simulations. For example, all of the better tuning approaches (approaches 1-3) are presented in the section called "**Modeling** of Advances System". The main results for better tuning approaches are presented in Figure 5, which is called "**Modeling** of ex-situ trained MLP classifier". Fig. 5b captions says "**Modeled** relative tuning error ...". So we think it is already clear from the paper. To answer the last part of the comments, we have preliminary evidence though much more effort is needed to perform systematic experimental study. This work hinges on development of (faster) experimental setup and is an ongoing work. We hope to share these results in the future papers.

Reviewers' Comments:

Reviewer #1:

Remarks to the Author:

For Q1, I'm good with response.

For Q2, it's good that the authors added a couple of sentences with the new citation [56] is good, but I'm not satisfied with the 1T1R related matter for Suppl. Table 3 and 4.

While the authors claimed this paper is not about VMM design but rather device technology, Suppl. Table 3 and 4 are actually about VMM and VMM-based MLP/accelerator benchmarking.

My point is that, there are many 1T1R silicon demonstrations in the literature for VMM, and many of them use commercial 1T1R binary devices that consume 10s of F^2 per 1T1R cell. Using these binary devices, one can implement multi-bit weights and multi-bit precision VMM, and I would say that is considered the baseline in the literature.

The 375 F^2 is supposedly needed for the analog-grade RRAM device, and this seems to be a worse-than-actual baseline that the authors generated, which benefits the comparison results in Suppl. Table 3 and 4.

The authors argued that if they use more aggressive ReRAM / high V_t assumptions, the overheads for local sensing and analog mux circuitry would reduce, but it can be seen from Suppl. Table 3 and 4 that "C: Analog mux" and "D: Local sensing circuit" have pretty small area from the overall 64x64 VMM block point of view.

The reason why the total 64x64 VMM area is smaller for the 0T1R schemes is because the 1T1R bitcell area that the authors assumed is too large. And again, for an analog-grade 1T1R device, this is kind of understood, but if we use multiple binary 1T1R devices, I hypothesize that the overall 64x64 VMM area of the 1T1R scheme could be smaller than the proposed 0T1R scheme.

One more thing: how did the authors estimate the level shifter area in Suppl. Table 3 and 4?

$11.5 \mu\text{m}^2 / 64 = 0.18 \mu\text{m}^2$ for one level shifter? This seems too small.

As you can see from a recent ISSCC 2021 paper - <https://ieeexplore.ieee.org/document/9365926> - Fig. 29.1.7 shows the level shifter area is quite large.

Reviewer #2:

Remarks to the Author:

I don't have any further question.

Reviewer #3:

Remarks to the Author:

The authors have addressed all my concerns. I would be happy to support the acceptance of this manuscript, which represents a significant step for large-scale memristor crossbar integration.

Response to Reviewers

The feedback from reviewers is enumerated for easy reference and is shown in black, while our responses are shown in blue.

Reviewer #1 (Remarks to the Author):

1. For Q1, I'm good with response.

Thank you!

2. For Q2, it's good that the authors added a couple of sentences with the new citation [56] is good, but I'm not satisfied with the 1T1R related matter for Suppl. Table 3 and 4.

It seems that there is a major confusion in interpretation of the Supplementary Table 3 (and possibly Supplementary Table 4) results – please see more on that in our replies to comments #5 and #7 below. The detailed information on how to interpret the reported breakdown data were already provided in footnotes 9 and 10. Just in case, we added more clarification in the table – see the text highlighted with yellow in Supplementary Table 3.

3. While the authors claimed this paper is not about VMM design but rather device technology, Suppl. Table 3 and 4 are actually about VMM and VMM-based MLP/accelerator benchmarking.

Supplementary Table 3 and Table 4 were not in the initially submitted paper and were added to the revised version in part to address the following Referee's comment "... *advantages of high-density passive crossbar circuit in application were not proved in this manuscript.*" As we mentioned in our previous response, we believe that such a performance study is an orthogonal line of research, not related to the main results of our paper, and in our opinion, would deserve publication on its own.

4. My point is that, there are many 1T1R silicon demonstrations in the literature for VMM, and many of them use commercial 1T1R binary devices that consume 10s of F^2 per 1T1R cell. Using these binary devices, one can implement multi-bit weights and multi-bit precision VMM, and I would say that is considered the baseline in the literature.

We believe that there are denser 1T1R commercial devices is not related is to the fact that such devices are binary because the switching physics is typically the same for binary and analog ReRAM devices. For example, joule heating is essential for switching memory states in the most practical (i.e., nonvolatile) metal-oxide ReRAMs. That puts a limit on how much currents and voltages can be reduced, and, therefore, in turn, sets a limit on the minimum size of an access transistor in a memory cell. The commercial processes with smaller cell sizes might be a result of using more refined industrial-grade fabrication technology (that, e.g., allow for better control of leakage currents and hence more efficient use of externally applied currents) and/or from using proprietary high-V process allowing to have more compact CMOS access transistors. As we mentioned in our previous response, our analog grade memory and VMM circuits would also **at least equally** benefit from these improvements (and in light of earlier misunderstanding of our results, we hope that Reviewer now would agree with that) and we performed such study – see Supplementary Figure 11, which shows that the gap between 1T1R and 0T1R becomes larger as the device currents are reduced. Moreover, we suspect that these dense commercial memories that Referee refers to are intrinsically analog, i.e., their states can be continuously tuned. However,

because of significant device variations, or the benefit of having more compact peripheral circuitry for the IP macroblocks, or other business-related issues, commercial foundries offer only “binary” functionality in such memories.

5. The 375 F^2 is supposedly needed for the analog-grade RRAM device, and this seems to be a worse-than-actual baseline that the authors generated, which benefits the comparison results in Suppl. Table 3 and 4.

The authors argued that if they use more aggressive ReRAM / high V_t assumptions, the overheads for local sensing and analog mux circuitry would reduce, but it can be seen from Suppl. Table 3 and 4 that “C: Analog mux” and “D: Local sensing circuit” have pretty small area from the overall 64×64 VMM block point of view.

We believe that the Referee misunderstood the numbers in Supplementary Table 3. “C: Analog mux” and “D: Local sensing circuit” represent the area (and energy) per **single** channel of the array. Note that a detailed equation on how the total area is calculated was already included in footnotes 9 and 10. In fact, the situation is precisely the opposite of what the Referee wrote, i.e., the total area in our 64×64 blocks **for all** studied 0T1R cases is **dominated** by a level shifter, analog muxes, and local sensing circuitry.

6. The reason why the total 64×64 VMM area is smaller for the 0T1R schemes is because the 1T1R bitcell area that the authors assumed is too large. And again, for an analog-grade 1T1R device, this is kind of understood, but if we use multiple binary 1T1R devices, I hypothesize that the overall 64×64 VMM area of the 1T1R scheme could be smaller than the proposed 0T1R scheme.

We believe that this is answered in our reply to comment #4. Because of a misunderstanding of the results of Supplementary Table 3, it seems that the Reviewer believes that the area of our 0T1R VMM design is dominated by the area of memory cells, which is not the case. As we mentioned in our previous response, we believe that the design based on multiple binary 1T1R or 0T1R devices would be inferior (in energy, area, or delay) to analog 1T1R or 0T1R devices.

7. One more thing: how did the authors estimate the level shifter area in Suppl. Table 3 and 4?

$11.5 \mu\text{m}^2 / 64 = 0.18 \mu\text{m}^2$ for one level shifter? This seems too small.

As you can see from a recent ISSCC 2021 paper -

<https://ieeexplore.ieee.org/document/9365926> - Fig. 29.1.7 shows the level shifter area is quite large.

This is related to the same confusion that we already addressed in our reply to comment #5. $11.5 \mu\text{m}^2$ is the area **per one channel**, so that in our estimates, we assumed the total area occupied by level shifters in 64×64 block is $11.5 \mu\text{m}^2 * 128$, i.e., 8,000 larger (!) than Referee seems to think. These area numbers are based on the actual layout in the 65 nm process. Note that the detailed equation was already provided in footnotes 9 and 10 in which we explained how the total area is computed.

Reviewers' Comments:

Reviewer #1:

Remarks to the Author:

Regarding Suppl. Table 3, thanks for pointing out that I missed the factor of 128 and 64 for rows B/C and D, respectively. Now the level shifter area per channel certainly makes sense.

I think the fact that the authors added the "per channel" phrase in the table itself will avoid confusion of the readers like me.

While I misunderstood the total area of rows B/C/D for Suppl. Table 3, my statement that "if we use multiple binary 1T1R devices (with 10s of F^2), the overall 64x64 VMM area of the 1T1R scheme will be much smaller" still seems to be true.

I agree with the authors that the commercial 1T1R cells are intrinsically analog, and they are used as binary devices due to variations, etc. But that's not my point. My point is that, besides the analog 1T1R device that the authors cited [16-19] from which the authors aggressively scaled and put in Suppl. Table 3, there are many commercial 1T1R devices that have been published and you can achieve ~analog functionality with multiple binary 1T1R devices with much smaller area. There are existing papers that achieved the same VMM functionality (with multiple binary 1T1R devices), so shouldn't these published results be the baseline for 1T1R in Suppl. Table 3, instead of a 1T1R device that is not fabricated and based on the authors' assumption?

But I also understand that maybe this is not related to the main results of this paper. I would finally suggest, in the footnote 1 of Suppl. Table 3, for the authors to add a brief statement about the fact that there are much denser commercial 1T1R cells, and the justification of why those are not used in this comparison.

I appreciate the authors' iterative responses on my nitpicky point.